# Yeast β-Glucans as Fish Immunomodulators: A Review

**DOI:** 10.3390/ani12162154

**Published:** 2022-08-22

**Authors:** Cristian Machuca, Yuniel Méndez-Martínez, Martha Reyes-Becerril, Carlos Angulo

**Affiliations:** 1Immunology & Vaccinology Group, Centro de Investigaciones Biológicas del Noroeste (CIBNOR), Instituto Politécnico Nacional 195, Playa Palo de Santa Rita Sur, La Paz 23096, Mexico; 2Facultad de Ciencias Pecuarias, Universidad Técnica Estatal de Quevedo (UTEQ), Quevedo 120301, Ecuador

**Keywords:** biomolecules, functional carbohydrates, immunity, infectious diseases

## Abstract

**Simple Summary:**

The β-glucan obtained from yeast—a very important molecule for fish production—activates the immune system of fish by different mechanisms and induces protection against pathogens. However, most previous related studies have focused on the use of commercial β-glucan from the yeast *Saccharomyces cerevisiae* to understand the activation pathways. Experimental β-glucans extracted from other yeasts show other interesting biological activities even at lower doses. This review article analyzes the current information and suggests perspectives on yeast β-glucans.

**Abstract:**

Administration of immunostimulants in fish is a preventive method to combat infections. A wide variety of these biological molecules exist, among which one of the yeast wall compounds stands out for its different biological activities. The β-glucan that forms the structural part of yeast is capable of generating immune activity in fish by cell receptor recognition. The most frequently used β-glucans for the study of mechanisms of action are those of commercial origin, with doses recommended by the manufacturer. Nevertheless, their immune activity is inefficient in some fish species, and increasing the dose may show adverse effects, including immunosuppression. Conversely, experimental β-glucans from other yeast species show different activities, such as antibacterial, antioxidant, healing, and stress tolerance properties. Therefore, this review analyses the most recent scientific reports on the use of yeast β-glucans in freshwater and marine fish.

## 1. Introduction

Given the accelerated growth of the aquaculture industry, the use of whole yeasts and their derived compounds as immunostimulants has been shown to be an excellent approach [1]. Yeasts are unicellular organisms distributed worldwide in a wide range of environments [2,3]. Their benefits are so extensive that they are also used in feed production as partial protein replacers [4,5]. Yeasts play a biological role within microbial communities in the fish intestine, including nutrient supply, pathogen control, and mucosal immunity maintenance [6,7]. Additionally, compounds of interest in the yeast cell wall promote biological activities in fish, such as mannan-oligosaccharide (immunostimulant) [8] and β-glucan (wound healing, stress resistance, immunostimulant, and disease protection) [9,10,11,12,13]. For instance, cell wall β-glucans have generated immunobiological activities in various animal taxonomic groups (birds, crustaceans, mammals and fish) [14,15,16,17]. Furthermore, β-glucans support other biological activities, including antibacterial [18], antioxidant [19], wound healing [11], and stress tolerance [12] effects. β-glucans are polysaccharides composed of glucose monomers joined by glycosidic bonds [20]. Their immunostimulant activities have been attributed to chemical composition, structural conformation, and molecular weight, among other factors [21]. All these characteristics depend on the yeast strain’s origin, and may affect their immunostimulant properties (Table 1). Meanwhile, β-glucans are recognized by several immune cell receptors [17,18] and generate immune responses that strengthen resistance to pathogenic bacteria, fungi, parasites, and viruses [11,19]. β-glucans have been shown to promote disease resistance by stimulating the immune system in fish species [12,21]. However, a possible immune signaling pathway, dose, and effective route of administration have not yet been indicated. Added to this is the potential of experimental β-glucans extracted from other yeasts, which can be used to benefit freshwater and marine fish production. Therefore, this review analysed recent information on the use of yeast β-glucans in fish, according to the following search patterns: “yeast β-glucans fish”, “recognition yeast β-glucans”, “immunomodulation of yeast β-glucans fish”. Relevant perspectives and the direction of future research are also discussed.

## 2. Yeast Cell Wall and β-Glucan Composition

### 2.1. Yeast Cell Wall

The yeast cell wall is a 100 to 150-nm thick cell armor (hard and rigid), representing approximately 15 to 32% of the dry weight [14,27] and 25 to 50% of the cell volume [15]. It is composed of approximately 85% polysaccharides (β-glucan and chitin) and 15% proteins (manno and transmembrane proteins) [16] (Figure 1). The yeast cell wall composition and organization form a layered ultrastructure that can be observed by electron microscopy [17,28].

### 2.2. Yeast β-Glucans

Yeast β-glucans are structured polysaccharides formed by monosaccharides (glucose), called “beta” (β) because of the specific glucosidic bonds (β-1,3 and 1,6) to which they are linked [29]. Due to the interaction of intermolecular polyhydroxyl groups, their structure can be single helix or triple helix, which makes them insoluble in water and in organic solvents (i.e., ethanol) [25]. This structural complexity endows β-glucans with high molecular weight that can vary according to the yeast species from which the β-glucan is extracted (Table 1).

Various studies have shown that insoluble β-glucans (β-1,3 and β-1,6) have superior capacity as biological response modifiers compared to soluble β-glucans (β-1,3 and β-1,4) [30]. Research reports have demonstrated that the most effective immunoenhancing activities, such as cell proliferation, were attributed to β-glucans with triple helix structures [24,30,31]. Observed effects included cell proliferation, phagocytic, antibacterial, and antioxidant activities, and immune-related gene expression [32,33].

## 3. Yeast β-Glucan Extraction

The extraction method has a significant influence on the physicochemical properties of β-glucans. Various methods have been proposed for β-glucan extraction, including physical [34,35,36], chemical [37,38], and enzymatic methods [39,40] (Figure 2). In the following, extraction methods using different processes are described, mainly based on cell disruption to release the cell contents and separate the β-glucan.

### 3.1. Physical Method

Physical disruption is a non-contact method that uses external force to achieve cell membrane rupture [38]. The different methods include sonication, homogenization, and bead milling. Among these methods, sonication has been among the most popular for obtaining β-glucans [36]. Cell wall disruption by sonication is caused by ultrasonic vibrations that produce a high-frequency sound, causing physical modifications that permit the solvent to penetrate into solids, increasing the diffusion rate of the desired molecule to the solvent [34]. Homogenization and bead milling lysis provide kinetic energy for cellular disruption and the release of intracellular components [36]. The latter method is old, and little used in research: cell disruption is prompted by hydraulic pressure (Frances press) and the method continues to be used in industry because of the low cost of operation. This method consists of applying direct pressure to release the intracellular contents [41].

### 3.2. Chemical Method 

Chemical cell lysis can be achieved by using specific chemicals to disrupt the cell wall, forcing it to release its contents [34]. This method is gentler than the physical approach and is suitable for lysing bacterial, fungal, and yeast samples [35]. Therefore, the chemical method is one of the most frequently employed to obtain β-glucans from yeast and other species [40]. Chemicals used for β-glucan extraction include alkali and/or acid organic solvents, such as acetic acid and sodium hydroxide, among others [42]. Organic solutions break down the cell wall through the difference in electronic charge, and also cause residues that contain chitin, glycogen, and proteins to be dropped [39]. It was recently reported that with this method a β-glucan of high quality, quantity, and biological activity was obtained.

### 3.3. Enzymatic Method

In recent years, biotechnological isolation methods with enzyme treatment have been developed [43]. Enzyme-based β-glucan extraction from yeast is a potential alternative to conventional solvent-based extraction methods, and possesses the advantages of being environmentally friendly, highly efficient, and a simplified process. Currently, enzymes including chitinase, proteases, and lipases have been widely used to degrade yeast cell walls and improve β-glucan isolation [39].

The final step of β-glucan extraction is purification, which consists of separating certain components found in yeast cell walls. In this sense, centrifugation and chromatography have been used for removing lipids and proteins from the cell wall, leading to more purified fractions of β-glucans [40].

## 4. Effects of Yeast β-Glucans on Fish Immune System

Fish have innate and adaptive immune systems. β-glucans are considered a type of pathogen-associated molecular pattern (PAMP) [44]. As such, they generate a signaling pathway in fish, but this has yet to be described in specific detail. The signaling pathway is described and exemplified in Figure 3, based on the latest research with β-glucans in fish.

After orally administration, β-glucans reach the intestine of the teleost fish; epithelial enterocytes synthesize apolipoprotein A-IV (apoa4) related to carbohydrate and lipid metabolism that probably captures β-glucan and secretes it into systemic circulation. Cytoplasmic actin 1 (actb) is permanently present in the intestinal microvilli, which together with transgelin (tagln) participate in actin-dependent β-glucan uptake [45]. Additionally, the presence of TLR-like receptors (Tlr2) in intestinal enterocytes could participle in the recognition of yeast β-glucans [46]. When β-glucans enter the systemic circulation, they are recognized by certain receptors, such as the three types of lectin C (a, b, and c) found in innate immune cells, and together with the spleen tyrosine kinase (Syk) generate intracellular signal transduction downstream by the mitogen-activated protein kinase (mapkin2) and nuclear factor kappa B (NF-κB) pathways [45,47,48]. Toll-like receptors (TLR 2/6) together with the myeloid differentiation primary response adapter protein 88 (myd88) also generate signaling cascades that cause activation of NF-κB [49]. The complement receptor (CR3) is a heterodimeric integrin that constitutes a critical link between cells and the extracellular matrix, functioning as anchoring sites and central elements for detection, processing, and transduction of the information received by β-glucans [50]. When NF-κB is activated, it initiates the expression of several pro- and anti-inflammatory cytokines [51]. Some of these cytokines have activities in the adaptive immune system, such as IL-6 and IL-10 that play important roles in the humoral immune response and induce differentiation of B lymphocytes [52,53]. IL-11 is another cytokine involved with anti-inflammatory characteristics, and is only characterized in a certain number of teleost fish [54]. When B lymphocytes recognize β-glucans, they begin to secrete immunoglobulins, such as IgM and IgT, involved in mucosal immunity [55]. IgT or IgZ is specific in teleost fish and is related to the intestinal mucosa [56,57,58]. Finally, yeast β-glucans could be involved in adaptive immune responses [59,60], but additional studies are required to better understand their signaling pathways in fish species. 

Currently, the majority of studies (in vitro and in vivo) have used commercial β-glucans, and very few have assessed experimentally extracted yeast β-glucans. Up to now, the main yeast strains for β-glucan extraction have been those belonging to *Saccharomyces cerevisiae*. However, non-*Saccharomyces cerevisiae* strains have also been tested with promising results, such as *Saccharomyces uvarum* [31], *Yarrowia lipolytica* N6 [61], *Sterigmatomyces halophilus* [18], *Debaryomyces hansenii* BCS004 [26], and *Cystobasidium benthicum* [22]. Remarkably, many studies have related β-glucan supplementation to increased disease resistance in fish. Therefore, the following sections describe the main outcomes obtained by the use of commercial and experimental β-glucans in freshwater and marine fish (Table 2, Table 3 and Table 4).

### 4.1. Freshwater Fish

Certain β-glucans used in freshwater fish aquaculture have been extracted from yeasts. MacroGard^®^ (Biotec-Pharmacon, TromsØ, Norway) is a commercial β-glucan extracted and purified from the cells of baker’s yeast *S. cerevisiae*, with a manufacturer’s recommended dose of 1 g/kg in the diet. At this inclusion rate it induces immune system responses and increases disease resistance in fish species including tench (*Tinca tinca*) treated with MacroGard^®^ (Biotec-Pharmacon, TromsØ, Norway) [63], as well as common carp (*Cyprinus carpio*), mirror carp (*Cyprinus carpio*), Nile tilapia (*Oreochromis niloticus*), and pacu (*Piaractus mesopotamicus*) treated with MacroGard^®^ (Biorigin, Sao Paulo, Brazil) [44,67,78,90], channel catfish (*Ictalurus punctatus*), rainbow trout (*Oncorhynchus mykiss*), and Nile tilapia treated with MacroGard^®^ (Biotec-Mackzymal, TromsØ, Norway) [59,64,68], rainbow trout treated with MacroGard^®^ (Biorigin, Scandinavia) [91], and striped snakehead fish (*Channa striata*) treated with MacroGard^®^ (no country of origin stated) [66].

Regarding timing, the optimal feeding regimen for Nile tilapia with the recommended dosage of MacroGard^®^ (Biotec-Mackzymal, TromsØ, Norway) is continuous administration for one week followed by one week of rest (“every-other-week”). By interrupting supplementation for two weeks, this feeding regimen activates the innate immune system and provides effective protection against pathogens (*Aeromonas hydrophila* and *Flavobacterium columnare*) [90]. In common carp, the same MacroGard^®^ (Biotec-Mackzymal, TromsØ, Norway) reduced the effects of intraperitoneal injections of lipopolysaccharides (LPS) and poly(I:C) or PAMPs, by activating the gene profile of different complementary system pathways [44]. Interestingly, in striped snakehead fish, MacroGard^®^ (no country of origin) at the recommended dose increased hematoimmune parameters, which were related to survival against *A. hydrophila* [66] (Table 2).

In other studies, higher doses of MacroGard^®^ induced better immune response, but may also have adverse effects. A dose of 2 g/kg MacroGard^®^ activates the expression of genes related to antioxidant activity, inflammation, stress, and immunity (innate and adaptive) within Nile tilapia with or without *Streptococcus iniae* challenge, and showed higher protective effect against pathogens compared to the recommended concentration [59]. A dose 2 g/kg of MacroGard^®^ in the diet of rainbow trout showed higher post-infection survival against *Yersinia ruckeri* than the recommended dose [68]. However, doses of 1, 2, and 5 g/kg of MacroGard^®^ in rainbow trout did not generate changes in cell subpopulations or humoral pre- and post-infection responses to *A. hydrophila*. Surprisingly, the 2 g/kg dose showed better expression patterns at 15 days in inflammatory genes, and at 30 days in those involved in physiological stress, while the post-infection expression was null. The dose of 5 g/kg had little effect on inflammatory gene expression pre- or post-infection. In contrast, the recommended dose not only prompted a post-infection inflammatory gene response but also increased expression of related genes in response to pre- and post-infection stress [91]. Consequently, prolonged β-glucan stimulation and high doses could suggest a generated immunosuppression event. Positive up-regulation of the genes involved in the physiological stress response may be indicative of stress axis desensitization to prolonged β-glucan stimulation.

In contrast, the immunosuppression effect did not occur in mirror carp after administering high doses (10–20 g/kg) of MacroGard^®^. The presence of an important infiltration of leukocytes into epithelial layer of the intestine was observed, without showing detrimental effects such as inflammation in the intestinal morphology. In addition, increases in monocyte proportions in peripheral blood were detected, presumably because monocytes are key components for replenishing macrophage and dendritic cell populations in the immediate response to inflammation sites [67]. Thus, an adequate dose of MacroGard^®^ is required, and a concentration of β-glucans below the optimal level may not be able to stimulate significantly the immune systems of the fish [63]. In view of several aspects that are still to be elucidated, including the optimum concentration, frequency, and duration, the species under study should also be taken into account. Because fish are physiologically diverse, their resistance to high glucan doses could be also different. 

In addition to MacroGard^®^, other commercial β-glucans have also been extracted from *S. cerevisiae*. These β-glucans have also induced immune responses that enhance resistance against pathogens, although the immunostimulation period is a critical factor for avoiding adverse effects. For example, continuous administration (56 days) of 5 g/kg of β-glucan (Angel Company, Wuhan, China) in the diet of koi carp caused immune fatigue [92]. In contrast, intraperitoneal administration of 10 µg/fish Yb-glucan (purity ≥ 98%, Sigma, St. Louis, MO, USA) induced immune responses (hematological, cellular, and humoral activity) and resistance to *Aeromonas veronii* infection [74]. 

Lately, many studies have focused on the use of β-glucans from new yeast sources that exhibit features including high yields or particular structural composition, enhancing immune activities in fish. For instance, β-glucan isolated from *Saccharomyces uvarum* administered at a dose of 10 g/kg induced cellular and humoral activity, and improved resistance to infection with *A. hydrophila* [31]. As with other β-glucans, its administration should not exceed 30 days to avoid leukocyte overstimulation which can limit sensitivity or cause tolerance to daily stimulation. In another study, β-glucan extracted from baker’s *S. cerevisiae* was administered intraperitoneally at a concentration of 10 mg/kg fish, to Nile tilapia three times at an interval of three days, which increased total leukocytes, phagocytic activity, and resistance to *A. hydrophila* infection [65]. 

Finally, in vitro experiments carried out outside the living organism, usually in tissues, organs, or cells, should be used as preliminary studies. This type of study enables confirmation of the safety of β-glucan extracted from yeast, and its immunostimulatory activities. Furthermore, such experiments help to elucidate the signaling pathway or immune activation at the cellular level, which is a very important point to consider in the study of new β-glucans that will later be experimentally used in vivo. For example, in vitro (10 μg/mL) and in vivo (5 µg/fish) stimulation by the commercial β-glucan Zymosan (Sigma, USA) administered intraperitoneally regulated the expression of pro- and anti-inflammatory genes and increased resistance to spring carp virus viremia (SVCV) in zebrafish (Danio rerio) [73]. In contrast, stimulation of 100 μg/mL of MacroGard^®^ (Biorigin, Sao Paulo, Brazil) and Zymosan (Sigma, St. Louis, MO, USA) in head kidney macrophages increased tlr2 gene transcription, indicating a potential recognition of β-glucan that generated oxidative activity by the production of reactive oxygen (ROS) and nitrogen (RNS) species. It also triggered the production of proinflammatory cytokines by increasing the expression of il-1b, il-6, and il-11 genes [47].

### 4.2. Marine Fish

As with freshwater fish, the commercial β-glucan MacroGard^®^ from *S. cerevisiae* has been the most frequently used to evaluate effects on the immune system and disease resistance of certain marine fish species, including Atlantic cod (*Gadus morhua L*.) with MacroGard^®^ (Biorigin Europe, Oslo, Norway) [83], Persian sturgeon (*Acipenser persicus*) with MacroGard^®^ (Biotec-Mackzymal, Tromsø, Norway) [84], white sturgeon (*Acipenser transmontanus*) with MacroGard^®^ (Biorigin, Sao Paulo, Brazil) [87], and Atlantic salmon (*Salmo salar*) and turbot (*Scophthalmus maximus*) with MacroGard^®^ (Biorigin, Lençois Paulista, Brazil) [45,85].

The recommended dose of MacroGard^®^ in the diet of Atlantic cod exerted an increase in proinflammatory gene expression in the foregut, hindgut, and rectum, in response to immersion bathing with the pathogen *Vibrio anguillarum* [83]. Although MacroGard^®^ is a good immunostimulant for oral administration, reports have indicated that further studies are needed to determine the optimal dose without adverse effects in marine fish. Surprisingly, doses higher than 1 g/kg generated different immune responses without adverse effects in two sturgeon species. In juvenile Persian sturgeon, doses from 2 to 3 g/kg MacroGard^®^ in the diet induced hematological responses and enhanced humoral immune activity by activating the alternative complement pathway, but the recommended dose did not generate a significant immune activity [84]. In white sturgeon, a MacroGard^®^ dose of 3 g/kg improved post-infection survival against the fungus *Veronaea botryosa*, an effect associated with the upregulation of proinflammatory genes and probably the enhancement of granulocyte/monocyte lineage cells [87].

After oral administration, β-glucan is neither digested nor absorbed in the intestine of the animals; however, it is recognized by superficial receptors of the leukocytes for as long as the dose administered orally can induce this response. According to the above, the 15 mg/kg dose of MacroGard^®^ by intubation in Atlantic salmon (*Salmo salar*) showed a localized uptake of β-glucans in the intestine, due to the abundance of goblet and immune cells. The impact of the molecules in the intestine induced metabolic activity by increasing the expression of the genes involved with carbohydrate, lipid, and energy metabolism, as well as β-glucan uptake. The pattern recognition expression was activated through transmembrane receptor genes of the lectin family with carbohydrate affinity and the complement receptor. Receptor activation initiated the expression of downstream immunoreceptor tyrosine-based activation (ITAM) motif signaling, in addition to proinflammatory and immunoglobulin gene expressions in the intestinal mucosa [45]. These results showed that a single dose administered by intubation was sufficient to induce different immune responses while maintaining immune homeostasis for several days, with no intestinal damage observed.

Seemingly, the degree of solubility is a key factor in β-glucan bioactivity for maintaining immune homeostasis. In this resepect, the intubation of Senegalese sole (*Solea senegalensis*) with 1 mg/fish of commercial insoluble β-glucan Yestimun^®^ (Quimivita, Barcelona, Spain) extracted from *S. cerevisiae* increased short-term proinflammatory expression and medium-term recognition by a receptor of the lectin group with carbohydrate affinity. This effect was related to a decrease in the relative bacterial proportion of the genus *Vibrio* in the intestinal microbiota, and could be used to modulate the population of the most popular taxonomic group in the gut microbiome of fish [93]. In contrast, in vitro soluble β-glucan (Biotec Pharmacon, Tromsø, Norway) at a dose of 100 μg/mL showed increases in gene transcription in Atlantic cod spleen cells, with bactericidal, proinflammatory, antioxidant, and glucose-metabolism-related activity [88]. Very likely the solubility of the molecule modulated the rapid responses, which could have a negative effect on the generation of long-lasting immune homeostasis. 

In marine fish, many studies have been conducted with the use of experimental β-glucans. In such studies, certain experimental β-glucans generated stronger immune responses than those obtained by commercial glucans. For instance, β-glucan from the marine yeast *Debaryomyces hansenii* BCS004 in Pacific snapper (*Lutjanus peru*) generated a cell proliferation effect, its inclusion in the diet (500 mg/kg) did not cause histopathological damage to the intestine, and it positively upregulated macrophage receptor genes. It also showed antioxidant properties that could help to reduce oxidative stress caused by ROS and RNS generated by different pathogens [26]. On the other hand, also in Pacific snapper, a dose of 200 μg/kg of β-glucan isolated from an extreme marine environment yeast (*Sterigmatomyces halophilus*) induced activity in vitro in different processes within phagocytic cells, and increased the transcription levels of anti- and pro-inflammatory genes. These activities were reflected in the inhibition of cytotoxicity caused by *A. hydrophila* challenges. Similarly, the β-glucan from the yeast *Cystobasidium benthicum* LR192 (Cb-βG) was proved to be a safe molecule in vitro after incubation in *Totoaba macdonaldi* thymus cells at three different doses (50, 100 and 200 μg/mL). It activated different oxidative processes in phagocytic cells, including production of ROS and RNS in the phagolysosome, and increased the mRNA levels of β-glucan receptor, macrophage differentiation and function, and proinflammatory cytokine genes [22]. Furthermore, mitochondrial ROS production could indicate a possible involvement of metabolic activity in response to immune cell activation. The effects shown by both experimental β-glucans indicate their potential as fish immunostimulants, and further study will help to elucidate their mechanisms of action. 

Finally, in an in vitro study, 50 μg/mL dose of Zymosan (Sigma, Z4250, St. Louis, MO, USA) strongly expressed activated T cell nuclear factor-c3 in Pacific snapper leukocytes. It also activated upstream and downstream immune-related gene expression [60]. Interestingly, a proportion of T lymphocytes were stimulated, so coactivity following stimulation with β-glucan is likely. This study leaves open the possibility of determining a potential involvement of yeast β-glucans in the adaptive immune systems of marine fish. 

Studies on the mechanisms of absorption by different pathways have received little attention in regard to marine teleosts; the available reports only discuss the beneficial biological effects of yeast β-glucans in fish. Therefore, studying the pharmacokinetics of these immunomodulatory substances is important to conclude these observations. Furthermore, the novel biological activities exhibited by β-glucans extracted from non-Saccharomyces yeasts can be exploited to study their potential for enhancing marine fish immunity.

## 5. Perspectives

Current reports have shown that most of the relevant research has focused on the use of commercial yeast β-glucans in fish to understand the mechanisms of innate immune action and disease protection. Therefore, in addition to addressing the problems of immunosuppression induced by high doses or prolonged periods of administration, future research should elucidate which β-glucans from yeast can modulate adaptive responses by their physicochemical characteristics. In addition, in-depth studies should be carried out focusing on activities including metabolic involvement by ROS generation in immune cells upon stimulation with yeast β-glucans. Undoubtedly the best candidates for these studies are experimental β-glucans that have been extracted and characterized from various yeast species. As shown in Table 2, Table 3 and Table 4, few studies have evaluated their effects on fish immune systems. 

Because of the wide distribution of yeasts, they can be isolated from multiple locations for the study of their β-glucans and the application of these in fish. Yeasts have been found in locations ranging from the aquatic [94], terrestrial [31], industrialized [95], and vegetable [96] even to the guts of various species such as birds [97], crustaceans [98], fish [99], and humans [100]. One particular yeast group is classified as extremophile because these yeasts are found in extreme environments, such as Antarctica [101], high-altitude UV-resistant volcanic areas [102], and high-temperature desert areas [103]. Some of these yeasts have probiotic characteristics due to their interactions with the digestive system and have potential for immune activation due to components found in their cell wall, including *Cystobasidium benthicum* (*Rhodotorula benthica*) [94], *Wickerhamomyces anomalus* [95], *Hanseniaspora opuntiae* and *Pichia kudriavzevii* [96], *Rhodosporidium paludigenum*, *Sporidiobolus pararoseus* and *Rhodotorula* sp. [98], *S. cerevisiae*, *Cryptococcus laurentii* and *Debaryomyces hansenii* [99], *Saccharomyces boulardii* [100], and *Rhodotorula mucilaginosa* [101].

Taking advantage of their wide distribution and probiotic characterization, species with potential for the extraction of β-glucans could be identified, relying on the methodologies already described for their isolation, extraction, and purification. Determinations of molecular weight, size, and structural composition are key elements for assessing immunogenicity effects. The biggest challenge will be to obtain β-glucans at industrial levels; after many years this has only been achieved with *S. cerevisiae*. Several chemical companies have well-established extraction processes that can obtain a product with a high degree of purification, for example, 90% β-1,3/1,6 from the Angel Yeast Co. (Wuhan, China) [13]. Although industrial and commercial extraction only occurs with bakery or brewery strains of *S. cerevisiae*, the possibility exists of identifying further species with good productive β-glucan yields.

The strong and prolonged immune response of yeast β-glucans is closely attributed to the molecular structural complexity of the 2–5 µm hollow and porous spheres [104]. Furthermore, β-glucans from other yeast species have been shown to generate stronger responses compared to commercially purified types. For example, in vitro β-glucan extracted from *S. halophilus* generated stronger immune responses and gene expression than Zymosan in Pacific snapper leukocytes after stimulation and infection with *A. hydrophila* [18]. The difference in immunostimulatory capacity could also be attributed to structure and molecular weight. For instance, β-glucan from *Debaryomyces hansenii* BCS004 measures 689.35 kDa [26] compared to 175 kDa (bakery) [23] and 240 kDa (brewery) for *S. cerevisiae* [25]. However, this is an unresolved hypothesis, as there are β-glucans with low molecular weight such as the kind extracted from the yeast *Cystobasidium benthicum* (2.32 kDa) that have proved to be a potential immunostimulant [22].

The study of yeast β-glucans has been important for some time; its history began in the 1940s with zymin (Zymosan) [105], starting with studies on its immunological effects and later for other applications [106]. Studies have demonstrated that MacroGard^®^ has improved wound healing in carp [107], rainbow trout [108] and silver catfish (*Rhamdia quelen*) [11], as well as resistance to low salinity stress in pompano (*Trachinotus ovatus* L.) [109]. Similarly, brewery *S. cerevisiae* β-glucans improved tolerance to ammonia stress in Mozambique tilapia (*Oreochromis mossambicus*) [12], and commercial β-glucan Angel provided protection against enteritis in rainbow trout [13] and was capable of reducing unpleasant odors generated during storage of silver carp meat (*Hypophthalmichthys molitrix*) [110].

In recent years, other newer applications of β-glucans have emerged based on structural manipulation to obtain nanoparticles. To our knowledge, no reports have yet been published on the use of β-glucan nanoparticles in fish, but similar studies are available from the agriculture, cosmetic and pharmaceutical industries [111,112,113,114]. Basically, β-glucan nanoparticles are small chains of low molecular weight (oligosaccharides), obtained after manipulating their original three-dimensional structure. They have exceptional advantages, including dose reduction due to a wide range of surface areas. β-glucan nanoparticles are attractive for future immunological studies in fish that remain to be performed. Undoubtedly, β-glucans from yeast comprise one of the most beneficial supplements for fish production.

## 6. Conclusions

Commercial yeast β-glucans have helped to enhance the immune defenses of fish. Interestingly, some yeast species have been found to contain β-glucans that are able to activate the immune system more efficiently than commercial ones. Therefore, new yeast species should be studied for physiochemical characterization and the evaluation of their β-glucans’ immunostimulatory effects on fish aquaculture.

## Figures and Tables

**Figure 1 animals-12-02154-f001:**
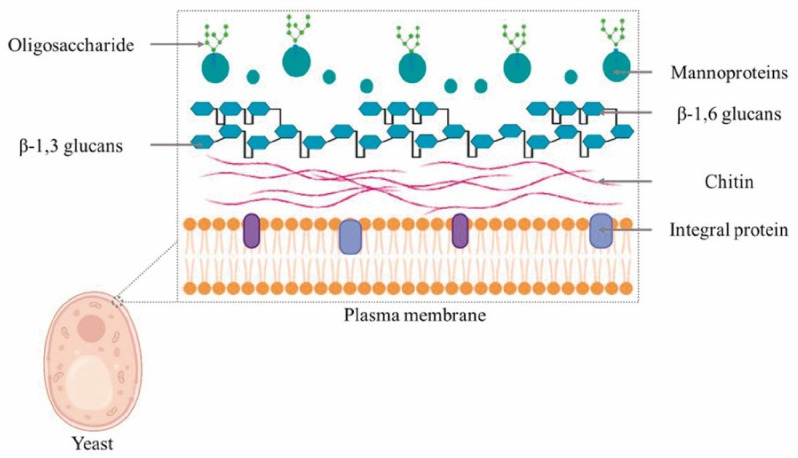
Conformation of the yeast cell wall. The cell wall is the largest, most resistant, and rigid organelle affecting the interaction with the external environment and the protection of the intracellular organelles, where compounds of great biotechnological interest are found including β-glucans.

**Figure 2 animals-12-02154-f002:**
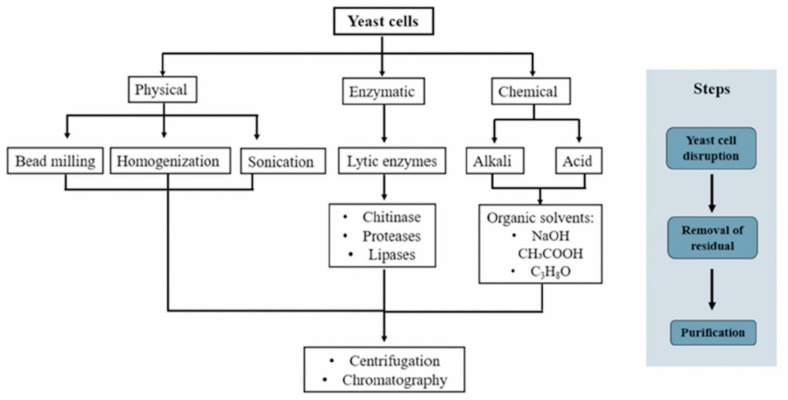
Representative scheme of the methods used for β-glucan extraction from yeast. Methodologies used for β-glucan extraction from yeast mainly differ in the method of breaking down the cell wall to release the internal components, and in the use of organic solvents to separate them. Finally, a purification process by centrifugation and chromatography is performed to obtain β-glucan from yeast.

**Figure 3 animals-12-02154-f003:**
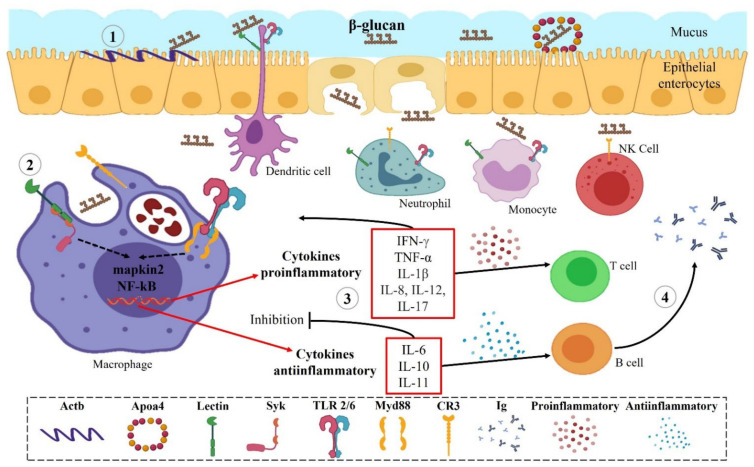
Signaling pathway for yeast β-glucans in teleost fish organisms. Proposed scheme of the β-glucan activation pathway in fish. (1) Intestinal epithelial enterocytes synthesize metabolic proteins activated by yeast β-glucan that secretes them into the systemic circulation. (2) Recognition of β-glucan by pathogen-associated molecular pattern receptors (PAMPs) that generate innate cellular immune responses and gene expression through translocation of the nuclear factor kappa beta (NF-κB) by phosphorylation, ubiquitination, and protein degradation. (3) Production of pro- and anti-inflammatory cytokines, receptors, and other proteins that activate the communication and activity of the adaptive immune system. (4) Production of immunoglobulins by B cells activated by the recognition of β-glucan.

**Table 1 animals-12-02154-t001:** Molecular weights of β-glucans from different yeast species.

Species	*Mw* *	Reference
*Cystobasidium benthicum*	2.32 kDa	[22]
*Saccharomyces cerevisiae* (bakery)	175 kDa	[23]
*Saccharomyces uvarum*	220 kDa	[24]
*Saccharomyces cerevisiae* (brewery)	240 kDa	[25]
*Debaryomyces hansenii* (BCS004)	689.35 kDa	[26]

* *Mw* = Molecular weight. kDa = KiloDaltons.

**Table 2 animals-12-02154-t002:** In vivo effects of yeast β-glucans on the immune systems of different species of freshwater fish.

Yeast Species (Origin)	Β-Glucan Type	Administration Dose and Route	Fish	Pathogen Challenge (Name, Dose, Route and Challenge Day)	Outcomes	Ref.
					(Relative Survival Upon Challenge and Increased Immune Parameters)	
*Saccharomyces uvarum* (β-glucan and whole cells)	β-1,3 y β-1,6	10 g Kg^−1^ Diet	*Cyprinus carpio*	*Aeromonas hydrophila* 1.5 × 10^6^ CFU mL^−1^	Survival: 77.8% and 71.6%	[31]
*S. cerevisiae* (bakery, Hang Zhou)	β-1,3 y β-1,6	60 days	*Oreochromis niloticus*	Intramuscularly 30 and 60 days	Significant increase in white blood cells, NBT, and serum lysozyme activity.	[62]
		21 days		Intra-peritoneal 21 day	Increase in cellular immunological parameters (neutrophil adhesion, macrophage oxidative oxide, lymphocyte transformation index, and phagocytic activity), and humoral parameters (bactericidal activity in serum, lysozyme and NO)	
*S. cerevisiae* (MacroGard^®^)	β-1,3 y β-1,6	0, 0.5, 1 and 2 g Kg^−1^ Diet	*Tinca tinca*	*Aeromonas hydrophila*	The 2 g Kg^−1^ dose had the lowest mortality after infection	[63]
		30 days		1 × 10^7^ CFU mL^−1^ Intraperitoneal at day 30	Additionally, increased respiratory burst activity in spleen macrophages, lysozyme activity, and total serum Ig levels	
*S. cerevisiae* (MacroGard^®^ and Betagard A^®^)	β-1,3 y β-1,6	1 g Kg^−1^ and 0.1 g Kg^−1^ Diet	*Ictalurus punctatus*	*Edwardsiella ictalurid* 9.5 × 10^6^ CFU mL^−1^	Survival: 56.7% and 46.4%	[64]
		7 and 14 days		Immersion 7 and 14 days	Increase in hematological parameters (% hematocrit, hemoglobin, TCC, RBC, WBC) and immunological parameters (SH50, lysozyme, total plasma protein)	
*S. cerevisiae* (bakery)	β-1,3 y β-1,6	10 mg Kg^−1^ fish Intraperitoneal	*Oreochromis niloticus*	*Aeromonas hydrophila* 1 × 10^6^ CFU mL^−1^	RPS: 83.3%	[65]
*S. cerevisiae* (MacroGard^®^)	β-1,3 y β-1,6	Nine days (injection every three days)	*Cyprinus carpio*	Intraperitoneal nine day	-	[44]
		15 days		Intraperitoneal Sampling 7 day	Increase in total leukocytes and phagocytic activity. Induced expression in CRP (crp1, crp2) and ACP (c1r/s, bf/c2, c3 and masp2)	
*S. cerevisiae* (MacroGard^®^)	β-1,3 y β-1,6	1 g Kg^−1^ Diet	*Channa striata*	*Aeromonas hydrophila* 1 × 10^7^ CFU mL^−1^	RPS: 61.54%	[66]
		84 days		Intraperitoneal 56 and 84 days 14 days mortality record	Increase of hematological parameters RBC, WBC, PCV, Hb%, VSG, serum protein, and immunological Ig and lysozyme activity	
*S. cerevisiae* (MacroGard^®^)	β-1,3 y β-1,6	10 and 20 g Kg^−1^ Diet	*Cyprinus carpio* L.	-		[67]
		56 days		-	Significant increase in localized infiltration of intestinal leukocytes, monocytes, and hematocrit value	
*S. cerevisiae* (MacroGard^®^)	β-1,3 y β-1,6	1–2 g Kg^−1^ Diet	*Oncorhynchus mykiss*	*Yersinia ruckeri* 2 × 10^8^ cells mL^−1^	RPS: Breeding females diet 2 g Kg^−1^ (42.2%) and fry diet 1 g/Kg (35.6%)	[68]
		90 days breeding females and 60 days fry		Immersion Sampling 25 days	Increased WBC, ACH-50, lysozyme activity, Ig, IgM	
*S. cerevisiae* (bakery)	β-1,3 y β-1,6	1 g Kg^−1^ Diet	*Pangasianodon hypophthalmus*	*Edwardsiella ictalurid* 8 × 10^4^ CFU mL^−1^	RPS: 37.7%.	[69]
		28 days		Immersion 28 day 14 days mortality record	Increased phagocytic activity, total IgM,	
*S. cerevisiae* (bakery)	β-1,3 y β-1,6	1 g Kg^−1^ Diet	*Pangasianodon hypophthalmus*	Edwardsiella ictaluri 1 × 10^6^ CFU mL^−1^	RPS: 83%.	[70]
		14 days		Immersion 14 day 24 h of infection	Overall expression of immune genes in the liver, kidney, and spleen	
*S. cerevisiae* (MacroGard^®^)	β-1,3 y β-1,6	1–2 g Kg^−1^ Diet	*Oreochromis niloticus*	*Streptoccus iniae* 2 × 10^7^ CFU mL^−1^	-	[59]
		21 days		Intraperitoneal Sampling one, three, and seven days	1 g kg^−1^: induced greater expression of the hsp-70, cxc chemokine, mhc-ii β and mx genes. Presented expression of hsp-70, mhc-ii β, and tlr 7 in the challenged group. 1 g Kg^−1^: induced expression of vtg, cas, igm-h, gst, il8, tnf-α in the unchallenged and challenged groups. More significant expression of hsp, cxc, and mhc-ii β in the challenged group.	
*S. cerevisiae* (MacroGard^®^)	β-1,3 y β-1,6	1 g Kg^−1^ Diet	*Brycon amazonicus*	*Aeromonas hydrophila* 3.8 × 10^8^ CFU mL^−1^	-	[71]
		15 days		Sampling 30 min and 24 h	Increased levels of cortisol, serum lysozyme, and complement system	
*S. cerevisiae* (MacroGard^®^)	β-1,3 y β-1,6	2 g Kg^−1^ Diet	*Channa striata*	*Aeromonas hydrophila* 2 × 10^6^ CFU mL^−1^	Resistance to bacterial infection.	[72]
		112 days immunization 56 days intake		Intraperitoneal 56, 120 and 168 days	Increase in hematological parameters (RBC, WBC, %PCV, Hb) and immunological parameters (Ig, lysozyme).	
*S. cerevisiae* (Zymosan)	β-1,3 y β-1,6	In vitro: 10 µg mL^−1^ ZF4 cells. In vivo: 5 µg fish Intraperitoneal	*Danio rerio*	Spring viremia of carp virus In vitro: 1 × 10^−3^ MOI In vivo: 10^4^ PFU mL^−1^	RPS: 59.7%	[73]
		In vitro:24 h In vivo:14 days		Immersion 14 day 17 days mortality record	Immunized and challenged + immunized fish showed increased expression of genes il-1b, il-6, il-8, il-10, and tnf-α	
*S. cerevisiae* (bakery, Sigma)	β-1,3 y β-1,6	10 μg fish Intraperitoneal injection	*Oreochromis niloticus*	Aeromonas veronii 1 × 10^6^ CFU mL^−1^	Relative survival 25%	[74]
		6, 12 and 24 h		Intraperitoneal 10 days mortality record	Increased hematological parameters. Cellular activity: lymphocytes, monocytes. Humoral activity: Total Ig, bactericidal activity, lysozyme, trypsin inhibition. Gene expression: tlr2, jak-1, nf-kb, il-1β, and tnf-1α.	
*S. cerevisiae* (bakery, Bettcan^TM^)	β-1,3 y β-1,6	1 g Kg^−1^ Diet	*Carassius auratus var. Pengze*	-	-	[75]
		70 days		-	Enhanced immunity and antioxidant capacity, increased acid phosphatase, alkaline phosphatase, glutathione peroxidase, reduced glutathione, catalase, and superoxide dismutase activities	
*S. cerevisiae* (MacroGard^®^)	β-1,3 y β-1,6	0.25 g Kg^−1^ Diet	*Cyprinus carpio*	*Aeromonas hydrophila* 5.01 × 10^8^ CFU mL^−1^	Survival > 50%	[76]
		63 days		Intraperitoneal 64 day 10 days mortality record	Increased lysozyme activity, complements and improves expression of immune genes (nk, lys, and il-8)	
*S. cerevisiae* (MacroGard^®^)	β-1,3 y β-1,6	1 g Kg^−1^ Diet	*Piaractus mesopotamicus*	*Aeromonas hydrophila* 1.5 × 10^8^ CFU mL^−1^ Inactivated at 50 °C	-	[77]
		15 days		Intraperitoneal Sampling 3 and 24 h	Increased plasma levels of cortisol, complement activity, and reduced numbers of monocytes and lymphocytes in peripheral blood	
*S. cerevisiae * (MacroGard^®^)	β-1,3 y β-1,6	5 g Kg^−1^ Diet	*Piaractus mesopotamicus*	*Aeromonas hydrophila * 1 × 10^2^ CFU mL^−1^	Increased cortisol, glucose, and CR3 y lysozyme by manipulation and bacterial inoculation.	[78]
		10 days		Intraperitoneal	Promoted inflammatory response in lymphocytes and neutrophils.	
*S. cerevisiae* (brewery Leiber^®^ Beta-S	β-1,3 y β-1,6	10 g Kg^−1^ Diet + *Lactobacillus plantarum* (1 × 10^8^ CFU cells mL^−1^)	*Rutilus rutilus*	-	Increased nonspecific humoral immunity parameters (lysozyme and total Ig)	[79]
		28 days		-	Cellular (pinocytic activity of phagocytes, respiratory burst)	
*S. cerevisiae* (bakery, Bettcan^TM^)	β-1,3 y β-1,6	2 g Kg^−1^ Diet	*Oncorhynchus mykiss*	*Aeromonas salmonicida* 3 × 10^5^ CFU mL^−1^	-	[49]
		42 days		Intraperitoneal Sampling four and six days	Differential expression of genes involved in immune or metabolic signaling pathways (fgg, fgb, f5, c9, c3, c5, tlr5, and myd88)	
*S. cerevisiae* (MacroGard^®^)	β-1,3 y β-1,6	0,1 g kg^−1^	*Oreochromis niloticus*	Aeromonas sobria and Streptococcus agalactiae	100% survival in immunized fish for 45 days	[80]
		15, 30 and 45 days		2 × 10^8^ and 1 × 10^8^ CFU mL^−1^ Intramuscular at day 10	Longer periods of administration of β-glucans increased growth, innate immune activity, and bacterial resistance	
*S. cerevisiae* [BY 4741 strain (G), MacroGard^®^ (M) and wild-type (W)]	β-1,3 y β-1,6	2 and 5 g Kg^−1^	*Oncorhynchus mykiss*	*Aeromonas salmonicida achromogenes*	G (2 and 5 g Kg^−1^) had the best survival rate	[81]
		15, 30 and 45 days		3.1 × 10^7^ UFC/100 g fish Intraperitoneal day 37	The G represented the best immunostimulant by increasing lysozyme activity, total Ig, and some immune genes (mcsfra, hepcidin) in the short and mid-term	
*S. cerevisiae* M/s Kuber	β-1,3 y β-1,6	5, 10 and 15 g kg^−1^	*Tor putitora*	*Aeromonas salmonicida*	RPS: 20% with diet 10 g kg^−1^	[82]
		56 days		2.5 × 10^7^ CFU mL^−1^ Immersion 56 day for 12 h 10 days mortality record	Total antioxidant levels increased, expression of cytokines such as tnf-α, il-1β, defensin1, c3 pre-post-challenge, and antiprotease activity increased only post-challenge	

**Table 3 animals-12-02154-t003:** In vivo effects of yeast β-glucans on the immune systems of different marine fish species.

Yeast Species (Origin)	β-Glucan Type	Administration Dose and Route	Fish	Pathogen Challenge (Name, Challenge Day, Dose and Route)	Outcomes	Ref.
					(Survival Upon Challenge and Increased Immune Parameters)	
*S. cerevisiae * (MacroGard^®^)	β-1,3 y β-1,6	1 g Kg^−1^ Diet	*Gadus morhua* L.	*Vibrio anguillarum* strain HI610 2.6 × 10^7^ CFU mL^−1^	-	[83]
		35 days		Immersion 36 day	Increased expression of anti-inflammatory genes (il-10 and ifn-γ). Active inflammation due to expression of pro-inflammatory cytokines (il1- β and il-8) post-challenge	
*S. cerevisiae* (bakery Fibosel ^®^)	β-1,3 y β-1,6	1 g Kg^−1^ Diet	*Lutjanus peru*	LPS 3 mg Kg^−1^	-	[19]
		42 days		Intraperitoneal	Improved growth, effectiveness in antioxidant enzymes (SOD and CAT) before and after exposure to LPS, activity of digestive enzymes (include trypsin, aminopeptidase, and chymotrypsin)	
*S. cerevisiae* (MacroGard^®^)	β-1,3 y β-1,6	1, 2 and 3 g Kg^−1^ Diet	*Acipenser persicus*	-	-	[84]
		42 days		-	Higher doses induced increases in WBC, %lymphocytes, and lysozyme and ACH-50 immune activity	
*S. cerevisiae* (MacroGard^®^)	β-1,3 y β-1,6	15 mg Kg^−1^ of fish	Atlantic salmon	-	-	[45]
		Sampling 1 and 7 days		-	Expression of β-glucan receptors sclra, sclrb, sclrc, and cr3; Syk, mapkin2, il1b, and mip2a target genes; apoa4 protein involved in carbohydrate metabolism; tagln, actb sensors	
*S. cerevisiae* (MacroGard^®^)	β-1,3 y β-1,6	0.5 g L^−1^ (incubated rotifers *B. plicatilis*)	*Scophthalmus maximus*	-	-	[85]
		10 days		-	Increase in chymotrypsin and trypsin activity. Complemented c3 activity and anti-inflammatory effect of hsp-70, tnf-α, and il-1β	
*S. cerevisiae* brewery (Yestimun^®^)	β-1,3 y β-1,6	1 mg/fish in PBS Oral intubation	*Solea senegalensis*	-	-	[86]
		sampling at 3, 24, 48 h and 7 days		-	Expression: il-1 β, clec, and irf7	
*Debaryomyces hansenii* BCS004	β-1,3 y β-1,6	500 mg Kg^−1^ Diet	*Lutjanus peru*	-		[26]
		28 days		-	Did not show pathological damages, edema, or inflammation in the intestine. Increased regulation of receptors (tlr2, dectin-2, c-type lectin-4, mmr-1)	
*S. cerevisiae* (MacroGard^®^)	β-1,3 y β-1,6	1 and 3 g Kg^−1^	*Acipenser transmontanus*	*Veronaea botryose*	RPS: 30% with diet 30 g Kg^−1^	[87]
		21 days		7.25 × 10^5^ spores mL^−1^ intramuscular	Increased expression of genes such as haptoglobin, serotransferrin, SAA, cathelicidin, and il-17, irf8 post-challenge	

**Table 4 animals-12-02154-t004:** In vitro effects of yeast β-glucans on the immune systems of freshwater and marine fish.

Yeast Species (Origin)	β-Glucan Type	Dose	Fish	Pathogen Challenge (Name, Dose and Route)	Outcomes	Ref.
					(Post-Challenge and Increased Immune Parameters)	
*S. cerevisiae* (bakery)	β-1,3 y β-1,6	100 μg/mL	*Gadus morhua*	-	-	[88]
				-	Increased antibacterial genes BPI/LBP and g-type lysozyme, pro-inflammatory cytokines il-1β and il-8, and antioxidants CAT and Cu/Zn-SOD	
*S. cerevisiae* (MacroGard^®^ and Zymosan)	β-1,3 y β-1,6	10–100 μg/mL	*Cyprinus carpio carpio*	-		[47]
				-	Increased production of reactive radicals (oxygen and nitrogen), expression of cytokine genes (il-1 β, il-6 and il-11)	
*S. cerevisiae* (Zymosan)	β-1,3 y β-1,6	50 μg/mL	*Lutjanus peru*	*Vibrio parahaemolyticus*	-	[60]
				1 × 10^8^ cell mL^−1^	Stimulated the expression upstream of ilf2, ilf3, can, and downstream of cd3, tcrβ, il-6, il-12	
*Yarrowia lipolytica* N6 (marine)	β-1,3 y β-1,6	200 μg/mL	*Lutjanus peru*	*Vibrio parahaemolyticus*	Immunized and challenged leukocytes	[89]
				1 × 10^8^ cell mL^−1^	Increased ON, SOD, CAT, PO. Regulated pro-inflammatory (il-1β, il-8, il-12, il-17) and anti-inflammatory (il-6, il-10) cytokines	
*Sterigmatomyces halophilus* (marine)	β-1,3 y β-1,6	200 μg/mL	*Lutjanus peru*	*Aeromonas hydrophila*	Immunized and challenged leukocytes	[18]
				1 × 10^8^ cell mL^−1^	Increased phagocytic activity, NBT, NO, PO, SOD, CAT. Genetic experimentation in cytokines il-1β, il-10, and il-17	
*Debaryomyces hansenii* BCS004	β-1,3 y β-1,6	100 μg/mL	*Lutjanus peru*	-	Increased cell viability with doses of 50, 100 and 500 μg/mL	[26]
				-	-	
*Cystobasidium benthicum*	β-1,3 y β-1,6	50, 100 and 200 μg/mL	*Totoaba macdonaldi*	-	Increased phagocytic activity, MPO, production of intracellular-mitochondrial ROS, NO, SOD, and gene expression of tlr2, clec17a, mmr, il-β, and 1csf1r2	[22]

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
