# Peer review of "Yeast β-Glucans as Fish Immunomodulators: A Review"

_animals, 2022, doi:10.3390/ani12162154_

Round 1

Reviewer 1 Report

This is a concise review for the immuno-modulatory effecit of beta-glucans on fish immunity. For clarity, the following issues should be addressed.

1. Extraction methodology is explained in some detail, but this seems bit out of scope of this review context, unless any functional relationship between immunomodulation and extraction methods.

2. Fig. 3 seems the most central information to be discussed in this review, but the information presented here is a mixture of knowledge from mammals and fish. In addition, the main text (Line 147-172) relevant to Fig. 3 contains citations of mammalian studies (#37-40, 44, 46, 47) and fish studies (#41-43, 48-56). Findings obtained from fish should clearly be discriminated from those from mammals to avoid such confusion.

L174-175: Saccaromyces cerevisiae is not a strain but a yeast species.

Some publication on non-Saccaromyces beta-glucans, such as screloglucan, should also be cited.

 https://doi.org/10.1111/j.1365-2761.1991.tb00613.x

 https://doi.org/10.1016/0044-8486(92)90023-E

Author Response

Reviewer 1

We thank the Reviewer for their valuable comments, and for allowing us to revise this manuscript and to improve it.  We have now addressed all suggestions in this revised version of the manuscript, and we hope that this version will now be acceptable for publication. Detailed responses to all suggestions are provided below.  All changes have been highlighted track mode revision in the manuscript for easy spotting.

This is a concise review for the immuno-modulatory effecit of beta-glucans on fish immunity. For clarity, the following issues should be addressed.

  1. Extraction methodology is explained in some detail, but this seems bit out of scope of this review context, unless any functional relationship between immunomodulation and extraction methods.

Replay: Thank you for your observation. It has been hypothesized that the extraction method by several authors could influence immune responses, therefore, we added sentence indicating this possibility with their respective references. We suggested to keep methodological strategies because they could be a useful guide for the readers. We hope you agree with us.

  1. Fig. 3 seems the most central information to be discussed in this review, but the information presented here is a mixture of knowledge from mammals and fish. In addition, the main text (Line 147-172) relevant to Fig. 3 contains citations of mammalian studies (#37-40, 44, 46, 47) and fish studies (#41-43, 48-56). Findings obtained from fish should clearly be discriminated from those from mammals to avoid such confusion.

Replay: Thank you very much for your comment. We have now described mechanisms of action specifically in fish, so the citations that included studies in mammals were removed and changed for those studies performed in fish.

L174-175: Saccaromyces cerevisiae is not a strain but a yeast species.

Replay: We corrected it.

Some publication on non-Saccaromyces beta-glucans, such as screloglucan, should also be cited.

 https://doi.org/10.1111/j.1365-2761.1991.tb00613.x

 https://doi.org/10.1016/0044-8486(92)90023-E

Replay: Thank you very much for your comment, but unfortunately, we would like not include scleroglucan because it has been obtained from the cell wall of a fungus and this review has only been focused on yeast β-glucans rather other sources of glucans.

Reviewer 2 Report

This is a good narrative review of available evidence on Beta glucan effects in fin fish aquaculture. There are many useful data provided (Table 2 for example) that make this review of use to researchers. By its nature though there is little attempt to critically evaluate contributing research (perhaps beyond the scope of this review?). Nonetheless, it does identify directions for future work. It would have be useful to see how your search was conducted? Without that, it is not easy to see how much of the field you have covered. I have made a number of suggestions about the writing and English expression (re-phrasing etc). Presumably the image quality will be re-assessed? Overall, a good and interesting read. Thank you.

Please see my suggested corrections / comments below:

Keywords – good

Line 28- this review analyses the most recent

37 – Yeasts play a biological role…

44 – joined by glycosidic bonds

47 – ‘origin’ is not needed

48 – awkward English expression and sentence too long?  “…immune cell receptors and generate immune responses that strengthen resistance…viruses. B-glucans have been shown to promote disease resistance by stimulating…”

51- analyses recent information on the use of…

59 – The yeast cell wall…of the dry weight…

Figure 1 – image quality?

80 – …endows B-glucans with…

84-86: immunoenhancing activities such as cell proliferation…were attributed to B-glucans with triple helix structures.

90 – delete ‘by’

105 – causing physical modifications that permit…

106 – penetrate solids?

107 – provide the KE for cellular disruption and the release of intracellular components.

112 – Therefore (not thereby)

115 – acetic acid and sodium hydroxide, among others.

116 – …and also cause the residues that contain chitin, glycogen and proteins, to be dropped…

125 – the last step of…

127 – “processes” – not required

131 – Replace ‘therefore’ with ‘As such’ or similar?

132 – The following signalling pathway is described below…? 132 – 134 needs re-phrasing.

152 – TLRs – write out toll like receptor in full (first mention)

173 – Currently, the majority of studies (in vitro and in vivo)   

179 – no need for “Thus”                                                                                                                            

 196 – 1g/kg in the diet. New sentence…At this inclusion rate, MacroGard is capable of…(reported capable of…?).

207 – how was protection provided? The examples cited – would like some critique of them. How good were the publications etc?

214/215 – induced better immune response…but “it may also generate” – it may also have generated…?)

205 and 217 – Nilotic tilapia or Nile tilapia - use one form?

224 – doesn’t make sense as its written? “in medium than in long terms”?

229 – delete ‘an’

234 – why is the leukocyte infiltration ‘important’?

236 – an increase in monocyte proportion, or, increases in monocyte proportions were detected, “presumably” because monocytes are key components in ….

239 – “an adequate dose”, or a threshold level, or an optimal level

241 – is the RT citation out of context? Should it be placed earlier in the discussion – around 225?

246 – does this paragraph repeat previous findings? Or could it be placed earlier in the discussion?

260 – delete bakery’s. Replace with baker’s

261 – and administered intraperitoneally at 10 mg/kg

266 – what is “it”? The in vitro studies?

267 – rephrase. Aforementioned problems to subsequently reflect then in an in vivo experiment? Do you mean in vitro is, in general, a better (easier?) study technique to identify issues that can subsequently be assessed / examined in in vivo experiments?

282 – immersion bath with the pathogen (not infectious bathing)

285 – doses higher than (not higher doses than)

292 – and probably the enhancement of

293 – passing the gastric barrier? It is still in the digestion process when it is in the intestines surely

309 – In this sense, the intubation of Senegalese sole (Solea senegalensis) with 1mg/fish of the commercial insoluble… (re-arrange the sentence)

314 – Because of the above? The Vibrio reduction? Re-phrase.

323 – In such studies

359 – delete ‘also’

365 – needs re-expressing. Aquatic, terrestrial, industrialised, vegetable, and even I the gut of various species such as birds, crustaceans, fish and humans

367 – because these yeasts

384 – ‘it’? – industrial / commercial production?

396 – starting with studies on its immunological but later, also on other applications

415 – beta-glucans have helped

Author Response

Reviewer 2

We thank the Reviewer for their valuable comments, and for allowing us to revise this manuscript and to improve it.  We have now addressed all suggestions in this revised version of the manuscript, and we hope that this version will now be acceptable for publication. Detailed responses to all suggestions are provided below.  All changes have been highlighted track mode revision in the manuscript for easy spotting.

This is a good narrative review of available evidence on Beta glucan effects in fin fish aquaculture. There are many useful data provided (Table 2 for example) that make this review of use to researchers. By its nature though there is little attempt to critically evaluate contributing research (perhaps beyond the scope of this review?). Nonetheless, it does identify directions for future work. It would have be useful to see how your search was conducted? Without that, it is not easy to see how much of the field you have covered. I have made a number of suggestions about the writing and English expression (re-phrasing etc). Presumably the image quality will be re-assessed? Overall, a good and interesting read. Thank you.

THANK YOUR FOR YOUR RECOMMENDATIONS. WE INCLUDED HOW WE SEARCH THE ARTICLES. PONER EN EL TEXTO QUÉ BUSCADORES Y QUÉ PALABRAS CLAVES SE USARON PARA IDENTIFICAR LOS ARTÍCULOS USADOS.

REVISAR SI SE PUEDE MEJORAR LA CALIDAD DE LAS IMAGENES Y AGREGARLAS.

Please see my suggested corrections / comments below:

TODAS ESTAS SUGERENCIAS SON DE INGLÉS, HAY QUE CORREGIRLAS EXACTAMENTE COMO INDICA.

Keywords – good

Line 28- this review analyses the most recent.

Replay: Done.

37 – Yeasts play a biological role…

Replay: Done

44 – joined by glycosidic bonds

Replay: Done

47 – ‘origin’ is not needed

Replay: Done

48 – awkward English expression and sentence too long?  “…immune cell receptors and generate immune responses that strengthen resistance…viruses. B-glucans have been shown to promote disease resistance by stimulating…”

Replay: Done

51- analyses recent information on the use of…

Replay: Done

59 – The yeast cell wall…of the dry weight…

Replay: Done

Figure 1 – image quality?

80 – …endows B-glucans with…

Replay: Done

84-86: immunoenhancing activities such as cell proliferation…were attributed to B-glucans with triple helix structures.

Replay: Done

90 – delete ‘by’

Replay: Done

105 – causing physical modifications that permit…       

Replay: Done

106 – penetrate solids?

Replay: Done

107 – provide the KE for cellular disruption and the release of intracellular components.

Replay: Done

112 – Therefore (not thereby)

Replay: Done

115 – acetic acid and sodium hydroxide, among others.

Replay: Done

116 – …and also cause the residues that contain chitin, glycogen and proteins, to be dropped…

Replay: Done

125 – the last step of…

Replay: Done

127 – “processes” – not required

Replay: Done

131 – Replace ‘therefore’ with ‘As such’ or similar?

Replay: Done

132 – The following signalling pathway is described below…? 132 – 134 needs re-phrasing.3

Replay: Done

152 – TLRs – write out toll like receptor in full (first mention)

173 – Currently, the majority of studies (in vitro and in vivo)    

Replay: Done

179 – no need for “Thus”        

Replay: Done                                                                                                                     

 196 – 1g/kg in the diet. New sentence…At this inclusion rate, MacroGard is capable of…(reported capable of…?).

Replay: Done

207 – how was protection provided? The examples cited – would like some critique of them. How good were the publications etc?

DESCRIBIR EN EL ARTÍCULO LAS DEBILIDADES DE LOS TRABAJOS Y QUE DEBERÍAN CONSIDERARSE PARA FUTUROS ESTUDIOS.  

Replay: Your suggestion was considered, and protection was induced by the administration regimen that activates the immune system and prolongs its activity even when the implementation was removed.

The articles were from impact and scientifically enhanced journals.

214/215 – induced better immune response…but “it may also generate” – it may also have generated…?)

Replay: Done

205 and 217 – Nilotic tilapia or Nile tilapia - use one form?

Replay: Nile tilapia is the correct form. It was corrected.

224 – doesn’t make sense as its written? “in medium than in long terms”?

Replay: Thank you very much for your comment. It was written correctly as described in the original article.

229 – delete ‘an’

Replay: Done

234 – why is the leukocyte infiltration ‘important’?

Replay: Done

It is the migration of leukocytes from the circulatory system to the intestinal tissue. It occurs mainly at sites of inflammation and the administration of high doses did not cause inflammatory damage.

236 – an increase in monocyte proportion, or, increases in monocyte proportions were detected, “presumably” because monocytes are key components in ….

Replay: Done

239 – “an adequate dose”, or a threshold level, or an optimal level

Replay: Done

241 – is the RT citation out of context? Should it be placed earlier in the discussion – around 225?

Replay: The citation has been removed as well as the text.

246 – does this paragraph repeat previous findings? Or could it be placed earlier in the discussion?

Replay: Repeated text was eliminated and the rest of the text was modified.

260 – delete bakery’s. Replace with baker’s

Replay: Done

261 – and administered intraperitoneally at 10 mg/kg

Replay: Done

Injected with 0.5 mL/fish at a concentration of 10 mg Kg-1

266 – what is “it”? The in vitro studies?

Replay: Done

267 – rephrase. Aforementioned problems to subsequently reflect then in an in vivo experiment? Do you mean in vitro is, in general, a better (easier?) study technique to identify issues that can subsequently be assessed / examined in in vivo experiments?

Replay: Done

In vitro experiments are very good options to determine the safety of experimental or extracted β-glucans from new yeast species. In addition, to evaluate their immunostimulatory potential to determine an effective concentration to then replicate in an in vivo experiment.

282 – immersion bath with the pathogen (not infectious bathing)

Replay: Done

It was performed as indicated.

285 – doses higher than (not higher doses than)

Replay: Done

292 – and probably the enhancement of

Replay: Done

293 – passing the gastric barrier? It is still in the digestion process when it is in the intestines surely

Replay: Done

Thank you very much for the observation, as the scientific report indicates, after oral administration, the β-glucan from yeast is neither digested nor absorbed.

309 – In this sense, the intubation of Senegalese sole (Solea senegalensis) with 1mg/fish of the commercial insoluble… (re-arrange the sentence)

Replay: Done

It was performed as indicated.

314 – Because of the above? The Vibrio reduction? Re-phrase.

Replay: Done

323 – In such studies

Replay: Done

359 – delete ‘also’

Replay: Done

365 – needs re-expressing. Aquatic, terrestrial, industrialised, vegetable, and even I the gut of various species such as birds, crustaceans, fish and humans

Replay: Done

367 – because these yeasts

Replay: Done

384 – ‘it’? – industrial / commercial production?

Replay: Done

396 – starting with studies on its immunological but later, also on other applications

Replay: Done

415 – beta-glucans have helped

Replay: Done

Reviewer 3 Report

In this review, the authors described "Yeast β-glucans as fish immunomodulators". Based on the research field, I don't think it's good enough for publication.

1. Yeast β-glucans as fish immunomodulators has been widely reported. The lights of this review?  

2. There are many Ms about the roles between β-glucans and intestinal microorganism. Since it's a review, the related studies should be added.

3. Why the studies of marine fish and fresh fish have been separated. What's the real difference.

4. Why authors deeply described the differential roles in different Yeast β-glucans?

Author Response

Reviewer 3

We thank the Reviewer for their valuable comments, and for allowing us to revise this manuscript and to improve it.  We have now addressed all suggestions in this revised version of the manuscript, and we hope that this version will now be acceptable for publication. Detailed responses to all suggestions are provided below.  All changes have been highlighted track mode revision in the manuscript for easy spotting.

In this review, the authors described "Yeast β-glucans as fish immunomodulators". Based on the research field, I don't think it's good enough for publication.

  1. Yeast β-glucans as fish immunomodulators has been widely reported. The lights of this review?  

Replay: Done

  1. There are many Ms about the roles between β-glucans and intestinal microorganism. Since it's a review, the related studies should be added.

Replay: Done

Yes, there are many studies where they describe the interactions of β-glucan in the gut of fish but specifically those extracted from yeast are already described in the review.

  1. Why the studies of marine fish and fresh fish have been separated. What's the real difference. 

Replay: They were separated by their physiological differences mainly linked to the aquatic environment. The salinity of freshwater is different from marine salinity, which influences the amounts of salts present inside the cells.

  1. Why authors deeply described the differential roles in different Yeast β-glucans?

Replay: Yeast β-glucans have been characterized as bioactive molecules and potential immunostimulants to improve fish health. But to date the immune activation pathway is not fully described so this review contributes to elucidating the possible mechanism of action based on studies reported in recent years. It also describes the immunostimulatory potential of β-glucans obtained from other yeasts. In addition, it indicates other bioactivities and recommends the direction that future studies should take for the benefit of fish farming.

Reviewer 4 Report

The work is good but requires some improvements.

Line 38-39, this statement is vague, detail.

Also mention which yeasts are used as an ingredient by the fish feed industry. For example, Candida utilis is highly used in the pet food industry as well as in fish feed. [1-4]

Line 82-83, this statement is also vague but extremely important, detail the diference in biological importance between insoluble and soluble β-glucans

3.1. Physical method

The French press methods should be stated, it's a rather old method not commonly used in research but still used in industry due to economical reason. The method involves high pressure the homogenate is then released directly in a cold recipient,  and due to the difference in pressure the cells explode immediately and the proteases have no time to react properly. It's an traditional method, very practical and cost effective. This should be mentioned in this chapter.

Little to nothing is said about β-glucan and ROS (reactive oxygen species), for example β-glucan on ROS production and energy metabolism in yellow croaker [5].

Line 246-247: state which other β-glucans products have been used.

Properly state the fish metabolic pathway involved.

1. Bzducha-Wróbel, A., BÅ‚ażejak, S., Molenda, M. et al. Biosynthesis of β(1,3)/(1,6)-glucans of cell wall of the yeast Candida utilis ATCC 9950 strains in the culture media supplemented with deproteinated potato juice water and glycerol. Eur Food Res Technol 240, 1023–1034 (2015). https://doi.org/10.1007/s00217-014-2406-6

2. Martin, A.M., Goddard, S. and Bemibster, P. (1993), Production of Candida utilis biomass as aquaculture feed. J. Sci. Food Agric., 61: 363-370. https://doi.org/10.1002/jsfa.2740610313

3. Reveco-Urzua FE, Hofossæter M, Rao Kovi M, Mydland LT, Ånestad R, Sørby R, Press CM, Lagos L, Øverland M. Candida utilis yeast as a functional protein source for Atlantic salmon (Salmo salar L.): Local intestinal tissue and plasma proteome responses. PLoS One. 2019 Dec 30;14(12):e0218360. doi: 10.1371/journal.pone.0218360. PMID: 31887112; PMCID: PMC6936787.

4.OLVERA-NOVOA, M.., MARTÍNEZ-PALACIOS, C.. and OLIVERA-CASTILLO, L.. (2002), Utilization of torula yeast (Candida utilis) as a protein source in diets for tilapia (Oreochromis mossambicus Peters) fry. Aquaculture Nutrition, 8: 257-264. https://doi.org/10.1046/j.1365-2095.2002.00215.x

5. Zeng, L., Wang, YH., Ai, CX. et al. Effects of β-glucan on ROS production and energy metabolism in yellow croaker (Pseudosciaena crocea) under acute hypoxic stress. Fish Physiol Biochem 42, 1395–1405 (2016). https://doi.org/10.1007/s10695-016-0227-1

Author Response

Reviewer 4

We thank the Reviewer for their valuable comments, and for allowing us to revise this manuscript and to improve it.  We have now addressed all suggestions in this revised version of the manuscript, and we hope that this version will now be acceptable for publication. Detailed responses to all suggestions are provided below.  All changes have been highlighted track mode revision in the manuscript for easy spotting.

The work is good but requires some improvements.

Line 38-39, this statement is vague, detail.

Replay: Done

Also mention which yeasts are used as an ingredient by the fish feed industry. For example, Candida utilis is highly used in the pet food industry as well as in fish feed. [1-4]

Replay: Done

Line 82-83, this statement is also vague but extremely important, detail the diference in biological importance between insoluble and soluble β-glucans

Replay: Done

3.1. Physical method

The French press methods should be stated, it's a rather old method not commonly used in research but still used in industry due to economical reason. The method involves high pressure the homogenate is then released directly in a cold recipient,  and due to the difference in pressure the cells explode immediately and the proteases have no time to react properly. It's an traditional method, very practical and cost effective. This should be mentioned in this chapter.

Replay: Done

Good method, in the study of Dallies & Paquet, (1998) obtained almost 100% cell rupture of cells after five passages in the press set at a pressure of 18,000 psi (21450 kg/cm2) with an optimum concentration of 50 mg/ml dry cell mass.

Little to nothing is said about β-glucan and ROS (reactive oxygen species), for example β-glucan on ROS production and energy metabolism in yellow croaker [5].

Replay: Thank you very much for your input, but it is impossible to cite that research because the β-glucan is obtained from barley.

Line 246-247: state which other β-glucans products have been used.

Replay: This section describes commercial β-glucans from yeast identified by different names corresponding to each commercial house and are not β-glucan products. We could consider the β-glucan nanoparticles mentioned in the section as a product.

Properly state the fish metabolic pathway involved.

Replay: Done.

  1. Bzducha-Wróbel, A., BÅ‚ażejak, S., Molenda, M. et al.Biosynthesis of β(1,3)/(1,6)-glucans of cell wall of the yeast Candida utilis ATCC 9950 strains in the culture media supplemented with deproteinated potato juice water and glycerol. Eur Food Res Technol 240, 1023–1034 (2015). https://doi.org/10.1007/s00217-014-2406-6
  2. Martin, A.M., Goddard, S. and Bemibster, P. (1993), Production of Candida utilis biomass as aquaculture feed. J. Sci. Food Agric., 61: 363-370. https://doi.org/10.1002/jsfa.2740610313
  3. Reveco-Urzua FE, Hofossæter M, Rao Kovi M, Mydland LT, Ånestad R, Sørby R, Press CM, Lagos L, Øverland M. Candida utilis yeast as a functional protein source for Atlantic salmon (Salmo salar L.): Local intestinal tissue and plasma proteome responses. PLoS One. 2019 Dec 30;14(12):e0218360. doi: 10.1371/journal.pone.0218360. PMID: 31887112; PMCID: PMC6936787.

4.OLVERA-NOVOA, M.., MARTÍNEZ-PALACIOS, C.. and OLIVERA-CASTILLO, L.. (2002), Utilization of torula yeast (Candida utilis) as a protein source in diets for tilapia (Oreochromis mossambicus Peters) fry. Aquaculture Nutrition, 8: 257-264. https://doi.org/10.1046/j.1365-2095.2002.00215.x

  1. Zeng, L., Wang, YH., Ai, CX. et al. Effects of β-glucan on ROS production and energy metabolism in yellow croaker (Pseudosciaena crocea) under acute hypoxic stress. Fish Physiol Biochem 42, 1395–1405 (2016). https://doi.org/10.1007/s10695-016-0227-1

Round 2

Reviewer 1 Report

Fig. 3 needs minor revisions as follows:

- The broken lines are used to show 'inhibition', but 'inhibition' is usually expressed by different sign such as ' |––––––––' by directing the flat head  to the inhibitory target.

- The interleukin (il) should be shown in upper case letters (IL-).

- 'K' of NF-KB should be latin letter 'kappa' in Symbol font.

- What do the pale and dense pink dots and blue dots near the T cell represent?

-

Author Response

Fig. 3 needs minor revisions as follows:

- The broken lines are used to show 'inhibition', but 'inhibition' is usually expressed by different sign such as ' |––––––––' by directing the flat head  to the inhibitory target.

Replay: Done, in the figure you can see the change.

- The interleukin (il) should be shown in upper case letters (IL-).

Replay: Done

- 'K' of NF-KB should be latin letter 'kappa' in Symbol font.

Replay: Done

- What do the pale and dense pink dots and blue dots near the T cell represent?

Replay: They refer to proinflammatory (pink-red dots) and anti-inflammatory (blue dots) cytokines. Now, they have been indicated into the figure.

Reviewer 3 Report

The revised Ms is good enough for being published

Author Response

The revised Ms is good enough for being published.

Replay: Thank you.

Reviewer 4 Report

Dear authors,

Thank you for improving your manuscript "Yeast β-glucans as fish immunomodulators: A review" which I now consider in a better state.

β-glucan is widely used in aquaculture to improve the immune system for a long time now, regardless of its source. As the work focuses on the most current information and suggests perspectives on yeast β-glucans, an aspects requires your attention:

1. "Little to nothing is said about β-glucan and ROS (reactive oxygen species), for example β-glucan on ROS production and energy metabolism in yellow croaker [5].

Replay: Thank you very much for your input, but it is impossible to cite that research because the β-glucan is obtained from barley."

Perhaps I should rephrase my statement.

Are β-glucans affecting ROS levels in fishes? As ROS part of the immune system and signaling cascade, as well as an indicator of toxicity.

How do β-glucans affect ROS (and why not asks ourselves RNS as well) in fish?

Author Response

Dear authors,

Thank you for improving your manuscript "Yeast β-glucans as fish immunomodulators: A review" which I now consider in a better state.

β-glucan is widely used in aquaculture to improve the immune system for a long time now, regardless of its source. As the work focuses on the most current information and suggests perspectives on yeast β-glucans, an aspects requires your attention:

  1. "Little to nothing is said aboutβ-glucan and ROS (reactive oxygen species), for exampleβ-glucan on ROS production and energy metabolism in yellow croaker [5].

Replay: Thank you very much for your input, but it is impossible to cite that research because the β-glucan is obtained from barley."

Perhaps I should rephrase my statement.

Are β-glucans affecting ROS levels in fishes? As ROS part of the immune system and signaling cascade, as well as an indicator of toxicity.

How do β-glucans affect ROS (and why not asks ourselves RNS as well) in fish?

Replay: Thank you for your explanation and now, information regarding β-glucans effects on ROS and RNS production has been incorporated as you suggested.